# Defeating Antibiotic-Resistant Bacteria: Exploring Alternative Therapies for a Post-Antibiotic Era

**DOI:** 10.3390/ijms21031061

**Published:** 2020-02-05

**Authors:** Chih-Hung Wang, Yi-Hsien Hsieh, Zachary M. Powers, Cheng-Yen Kao

**Affiliations:** 1Department of Power Mechanical Engineering, National Tsing Hua University, Hsinchu 30013, Taiwan; biowang2007@gmail.com; 2Institute of Biochemistry, Microbiology and Immunology, Chung Shan Medical University, Taichung 40201, Taiwan; hyhsien@csmu.edu.tw; 3Department of Medical Microbiology and Immunology, University of Wisconsin-Madison, Madison, WI 53706, USA; zmpowers@wisc.edu; 4Institute of Microbiology and Immunology, School of Life Science, National Yang-Ming University, Taipei 11221, Taiwan

**Keywords:** antibiotic resistance, alternative therapies, regulator, signal transduction, virulence factors

## Abstract

Antibiotics are one of the greatest medical advances of the 20th century, however, they are quickly becoming useless due to antibiotic resistance that has been augmented by poor antibiotic stewardship and a void in novel antibiotic discovery. Few novel classes of antibiotics have been discovered since 1960, and the pipeline of antibiotics under development is limited. We therefore are heading for a post-antibiotic era in which common infections become untreatable and once again deadly. There is thus an emergent need for both novel classes of antibiotics and novel approaches to treatment, including the repurposing of existing drugs or preclinical compounds and expanded implementation of combination therapies. In this review, we highlight to utilize alternative drug targets/therapies such as combinational therapy, anti-regulator, anti-signal transduction, anti-virulence, anti-toxin, engineered bacteriophages, and microbiome, to defeat antibiotic-resistant bacteria.

## 1. Introduction

With the global misuse of antibiotics, the rapid emergence and dissemination of antibiotic-resistant bacteria has been increasing yearly [1]. Therefore, antibiotic resistance is one of the most serious global public health threats of the 21st century. The World Health Organization has highlighted that antibiotic resistance in clinical treatment, if left to persist at the current rate, may lead to 10 million deaths and a reduction of 2–3.5% gross domestic product annually by 2050 [2,3]. Among antibiotic resistant bacteria, “ESKAPE” pathogens, including *Enterococcus faecium*, *Staphylococcus aureus*, *Klebsiella pneumoniae*, *Acinetobacter baumannii*, *Pseudomonas aeruginosa*, and *Enterobacter* species [4], cause the majority of hospital infections with higher mortality of patients. “ESKAPE” pathogens effectively escape the effects of commonly used antibacterial drugs, and are usually associated with significantly higher economic burden by increasing the duration of hospital stays and deceasing workforce productivity [3,5]. Recently, a notable study reported that ESKAPE pathogens represented 42.2% of species isolated from bloodstream infections in the United States [5]. Moreover, compared with patients infected with non-ESKAPE pathogens, patients presenting ESPAKE-bloodstream infections were associated with a 3.3-day increase in length of hospital stay, and a 2.1% absolute increase in mortality [5].

It has become critically challenging for clinicians to treat patients infected with multidrug resistant (MDR), extensively resistant (XDR), or pandrug resistant (PDR) bacteria. MDR bacteria are labelled for their resistance to more than one antimicrobial agent based on susceptibility tests in vitro [6]. In contract, PDR bacteria are resistant to all clinically relevant antimicrobial agents [6]. Patients infected with antibiotic-resistant bacteria acquire delayed antimicrobial therapy, the chance of successful clinical treatment deteriorates regardless of the origin of the patient or bacterial species [7,8].

Antibiotic resistance can be achieved by mutations in different chromosomal loci or horizontal acquisition of resistance genes (by plasmids, integrons, or transposons), with the greatest concern placed on the bacteria that have acquired transferable antibiotic resistance determinants [9,10,11]. Many strategies against antibiotics in bacteria have been reported, such as enzyme inactivation, changing cell permeability, altering target binding sites, increasing antibiotic efflux, and performing complex phenotypes changes (ex, biofilm formation) [9,10,11]. Carbapenems previously have been considered as the most effective broad-spectrum β-lactam antibiotics for the defense of MDR Gram-negative bacteria. Recently, carbapenem-resistant bacteria have emerged due to the resistance mechanisms described above and thus colistin (polymyxin E) and tigecycline are two antibiotics now considered as the “last resort” for treatment of carbapenem-resistant bacteria. However, concurrent with the increasing consumption of these two drugs, there are increasingly reports of colistin- or tigecycline-resistant bacteria within the last 5 years [12,13,14]. Additionally, infection with colistin or tigecycline resistant *K. pneumoniae* has been associated with increased risk hazard for in-hospital mortality [15,16].

Most identified colistin resistance mechanisms in Gram-negative bacteria involve changes to the lipopolysaccharide (LPS) structure, as colistin initially interacts with the negatively charged lipid A of LPS [17,18]. Importantly, plasmid-borne phosphoethanolamine transferases (*mcr-1* to *mcr-8*) have recently been identified and these plasmids threaten to increase the rate of dissemination of clinically relevant colistin resistance in CRE [18]. Bacteria confer resistance to tigecycline by the overexpression of *ramA* (a positive regulator of the AcrAB efflux system) or tigecycline-specific active efflux pump (*tet* variants), [12−14], [19,20]. Therefore, there is urgent need to develop novel therapies and classes of antimicrobials to fight bacterial infections [21].

A novel antibacterial drug is defined by the following criteria [22,23]: (1) belongs to a novel chemical class and interacts with a new target, (2) works via a new mechanisms or binding to new target sites, and/or (3) is biochemically modified to resensitize a previously resistant pathogen. Few novel classes of antibiotics have been discovered since 1960, and limited pipelines of new agents are under development. Moreover, once a new drug is introduced to the clinic, antibiotic resistance can arise rapidly via strong selective pressure soon after induction. In the past ten years, several targets for non-antibiotic therapies were developed for a post-antibiotic era. Here, we highlight the alternative drug targets and therapies to defeat antibiotic-resistant bacteria.

## 2. Combination Therapy I: Increases Membrane Permeability

Polymyxins are lipopeptide antibiotics with bactericidal activity against Gram-negative bacteria that work by disrupting the cell membrane via both hydrophobic and electrostatic interactions [17]. Tran et al. screened FDA approved drugs to identify potential synergistic candidates with polymyxins for MDR Gram-negative bacteria eradication [24]. They identified a non-antibiotic drug-mitotane (steroidogenesis inhibitor and cytostatic antineoplastic medication) as a potential candidate for combination therapy with polymyxin B against Gram-negative bacteria such as carbapenem-resistant *P. aeruginosa*, *A. baumannii*, and *K. pneumoniae* (Table 1) [24]. It is hypothesized that increased permeability of the outer membrane caused by polymyxin B may lead to the entry of mitotane into the bacterial cells, however, the mechanism by which mitotane inhibits the bacterial pathogen growth remains unclear. Several polymyxin derivatives have been developed to serve as antibiotic adjuvants have recently passed Phase 1 clinical studies. One of these drugs, SPR741, lacks direct antibacterial activity but disrupts the bacterial outer membrane, permeabilizing it to antibiotics (Figure 1A & Table 1) [25,26]. SPR741 exhibits potentiating antibiotic activity and extends the spectrum of activity of different antibiotics with diverse targets, thus effectively acting against antibiotic-resistant *E. coli*, *K. pneumoniae*, and *A. baumannii* in vitro [25,26,27,28]. Moreover, a murine-infection model demonstrated that treatment with SPR741 in combination with rifampin dramatically increased the survival rate of mice receiving the dual therapy over mice receiving SPR741- or rifampicin- treatment alone [26]. Loss of porin (outer bacterial membrane protein) restricts the influx of drug to the periplasm, and thereby enhances antibiotic resistance [29,30]. For example, mutations in porins OmpC/OmpF and OmpK35/OmpK36 have often been identified in carbapenem-resistant *E. coli* and *K. pneumoniae*, respectively [29,30]. However, whether SPR741 can increase the influx of carbapenems to the periplasm in porin-deficient carbapenems resistant bacteria is still unknown. Therefore, it is worth investigating whether combine SPR741 with carbapenem can be used for the treatment of porin-deficient carbapenem-resistant bacteria infection.

## 3. Combination Therapy II: Reduces Efflux Pump Activity

Efflux pumps expel antimicrobials from bacterial cells which may result in resistance to a wide spectrum of antibiotics as well as disinfectants [41,42]. Five major families of efflux transporters are identified in bacteria: the adenosine triphosphate (ATP)-binding cassette (ABC) superfamily, the major facilitator superfamily (MFS), the multidrug and toxic compound extrusion (MATE) family, the resistance-nodulation-division (RND) family, and the small multidrug resistance (SMR) family [43]. These five families have been defined on the basis of their sequence similarity, substrate specificity, energy source, number of components, and number of transmembrane-spanning regions [43]. Most of the MDR efflux pumps identified in drug-resistant bacteria are members of RND family [44,45].

A number of potent efflux pump inhibitors (EPIs) against antibiotic-resistant Gram-negative bacteria were reported, including 1-(1-naphthylmethyl) piperazine (NMP) [31], carbonyl cyanide *m*-chlorophenylhydrazone (CCCP) [46], phenylalanyl arginyl β-naphthylamide (PAβN, also called MC207,110) (Table 1) [31], and quinoline derivatives [47]. The mechanism of action of EPIs is through competitive inhibition where these inhibitors become the substrate of efflux pumps instead of the target antibiotics and thus the concentration of antibiotic increases intracellularly, eventually leading to cell death. Among EPIs, PAβN have been used as broad spectrum EPI for *P. aeruginosa* [32]. In addition to inhibition of efflux pump activity, PAβN permeabilizes bacterial membranes, decreases the level of intrinsic resistance significantly, and reduces frequency of emergence of fluoroquinolones- resistant *P. aeruginosa* strains [32,48].

Previous work demonstrated that some plant extracts and phytochemical products could be used as potentiators or synergists of antibacterial agents. For example, Negi et al. isolated curcumin, a nature extract derived from *Curcuma longa*, which acts as a permeabilizer and inhibits efflux pump systems in *P. aeruginosa* (Table 1) [33]. The other two natural products, EA-371α and EA-371δ, were identified through screening of a library of 78,000 microbial fermentation extracts [49]. EA-371α and EA-371δ were isolated from a strain closely related to *Streptomyces velosus*, and were shown to be specific and potent *P. aeruginosa* MexA-OprM efflux pump inhibitors [49]. However, no EPI has been clinically approved to date, mainly due to the toxicity problems, low in vivo efficacy, or poor pharmacokinetic properties [42,50,51].

## 4. Combination Therapy III: Inhibits Kinase Activity and Intrinsic Antibiotic Resistance

Two-component signal (TCS) transduction systems are stimulus-response coupling devices that allow bacteria to sense and adapt to environmental changes, including the challenges that pathogenic bacteria face inside the host (such as intracellular oxidative stress or gastric acid in the stomach) [52,53,54]. The bacterial two-component system is usually assembled with a sensor and a response regulator [54]. These play an important role in bacterial physiological function, including antibiotic resistance, biofilm formation, virulence, and cell division [52,55,56]. In *Salmonella enterica*, the PhoP/PhoQ two-component system controls the pathogenicity of the organism to establish infection in the host [57]. Carabajal et al. identified a series of quinazoline compounds that showed selective and potent down regulation of PhoP/PhoQ-activated genes by screening of 686 compounds from the GlaxoSmithKline (GSK) Published Kinase Inhibitor Set (Table 1) [34]. These quinazoline compounds are noncytotoxic and exhibit anti-virulence effect ex vivo by blocking *S.*
*Typhimurium* intramacrophage replication [34].

In 1991, a bacterial serine/threonine kinase with high structural homology to the eukaryotic protein kinase was first identified in *Myxococcus xanthus* [58]. Broad genomic studies revealed that eukaryotic-like serine/threonine kinases (eSTKs) were found to be nearly ubiquitous across bacterial species. Furthermore, many Gram-positive pathogens contain transmembrane eSTKs known as penicillin-binding-proteins and Ser/Thr kinase-associated (PASTA) kinases, which have been shown to regulate biofilm formation [59], cell wall homeostasis [60], metabolism [61], and virulence [62,63]. *Listeria monocytogenes* mutants deficient in the PASTA kinase, PrkA, show impaired growth under nutrient-limiting conditions and reduced survival and replication in host cells when compared to the wild-type strain [63]. Moreover, deletion of homologous PASTA kinase in some species, for example *L. monocytogenes* and *S. aureus*, has been linked to increased susceptibility to β-lactam antibiotics [64,65,66]. This contrasts the homologous *Mycobacterium tuberculosis* PASTA kinase, PknB, which is essential for survival [67]. Taken together, these results suggest that PASTA kinases may have potential as alternative targets in a variety of clinical pathogens.

Schaenzer et al. performed a small-molecule screening and identified GSK690693, an imidazopyridine aminofurazan-type kinase inhibitor that can dramatically increase the sensitivity of the intracellular pathogen *L. monocytogenes* to various β-lactams through inhibiting the activity of PrkA (Figure 1B) [66]. Moreover, GSK690693 potently inhibits PrkA kinase activity and exhibits significant selectivity for PrkA relative to the *S. aureus* PASTA kinase Stk1 [66]. Currently, pyrazolopyridazine GW779439X was found to resensitize methicillin-resistant *S. aureus* (MRSA) to various β-lactams through inhibition of the Stk1 (Figure 1B) [35]. Schaenzer et al. found that 5μM GW779439X were effective in lowering the minimal inhibitory concentration (MIC, μg/mL) of β-lactams oxacillin (16-fold, MIC 16 to 1) and nafcillin (8-fold, MIC 16 to 2) against MRSA LAC strain (Table 1) [35]. PASTA kinases are therefore considered attractive antibacterial targets in the future.

## 5. Anti-Regulators

*Vibrio cholerae* is the aquatic Gram-negative bacterium responsible for the human disease cholera. The two main virulence factors involved in *V. cholerae* pathogenesis are cholera toxin (CT) and toxin-coregulated pilus (TCP). CT is an ADP-ribosylating toxin composed of two subunits which lead to an increase in cAMP in intestinal cells, causing diarrhea due to the osmotic imbalance [68]. TCP is a type IV bundle-forming pilus that is involved in *V. cholerae* intestinal colonization [69]. The expression of CT and TCP is regulated by the master regulator, ToxT [70]. Hung et al. identified virstatin (4-[*N*-(1,8-naphthalimide)]-*n*-butyric acid) inhibits ToxT dimerization and thus reduces colonization of *V. cholera* in a murine model of infection (Figure 1C) (Table 1) [36,37]. Another small molecule inhibitor, toxTazin, blocks the production of an activator (TcpP) necessary for expression of the *toxT* gene and thus reduces the virulence of *V. cholera* (Figure 1C) [71]. 

## 6. Anti-Quorum Sensing System

Quorum sensing (QS) is a bacterial cell-cell communication process. Autoinducers (AIs) are QS signaling molecules that are produced in response to changes in cell-population density [72]. After colonization, bacterial population density increases, AIs accumulate in the microenvironment, and bacterial cells detect these messages to track changes in population size as a mechanism to modulate gene expression [72]. QS has been shown to control bacterial bioluminescence [73], biofilm formation [74], toxin secretion [75], motility [76], sporulation [77], and virulence factor expression [78]. This unique bacterial communication makes anti-QS compounds a promising way to selectively counter bacterial virulence, making the interference of QS and AIs production putative targets for alternative therapies.

A key player in *E. coli* QS is the sensor kinase QseC, which responds to the host stress hormones epinephrine and norepinephrine, as well as to the bacterial signal AI-3 [79]. QseC autophosphorylates when sensing its signals and then transfers its phosphate to downstream response regulators (RRs) such as QseB, QseF, and KdpE in Enterohemorrhagic *E. coli* (EHEC) [80]. Upon phosphorylation, these RRs directly regulate expression of different sets of genes in EHEC [80]. Homologous QS genes are encoded by *S. typhimurium* (87% similarity to EHEC QseC) and *Francisella tularensis* (57% similarity), both of which control virulence gene expression in the respective species [81,82].

A small molecule, LED209 shows activity to inhibit the binding of signals to QseC, and thus prevents its autophosphorylation, consequently inhibiting QseC-mediated activation of virulence factor expression (Figure 1D) (Table 1) [38]. LED209 abolishes EHEC attaching and effacing (A/E) lesion formation on cultured epithelial cells and decreases the expression of the *stxAB* genes that encode shiga toxin [32]. Importantly, LED209 does not directly inhibit the growth of pathogen. LED209 inhibits the virulence of EHEC, *S. typhimurium* and *F. tularensis* in vitro and in vivo [38]. 

Amino-terminated poly(amidoamine) (PAMAM) dendrimers have shown potential to function as a broad antibacterial agent (Figure 1D). PAMAM dendrimers inhibit the growth of both antibiotic sensitive and resistant Gram-negative bacteria significantly by destroying their cell membranes [83]. However, a critical concern for PAMAM is cytotoxicity to mammalian cells, which hampers PAMAM to be developed as a systemic antibacterial agent in the future [84]. Surprisingly, LED209 conjugated with PAMAM serve as a multifunctional agent, showing higher antibacterial activity against Gram-negative bacteria and lower cytotoxicity to host cells [85].

*P. aeruginosa* has emerged as a significant pathogenic bacteria in the nosocomial infections due to its resistance to commonly used antibiotics and harsh environments by modulating its highly organized QS and associated biofilm formation [86]. Pattnaik et al. showed the potential of *Diaporthe phaseolorum* SSP12 extract against *P. aeruginosa* PAO1 QS [87]. The in silico molecular docking results suggest that the bioactive metabolites DTBP (2,4-Di-tert butyl phenol) and fenaclon compete with the natural AIs to bind with the cognate RRs, RhlR and LasR, respectively, and thus reduce the production of biofilm and virulence factors in *P. aeruginosa* PAO1 [87]. However, the in vivo study is required to be performed to demonstrate the potential application of fanaclon and DTBP towards development of anti-*P. aeruginosa* therapies. 

## 7. Anti-Bacterial Virulence Factors (Adhesion and Motility)

After entering the host, bacterial motility mediated by flagella has been shown to be critical for bacterial trafficking to target cells/tissues, where the pathogen then utilizes adhesins to interact with appropriate receptors on host cells for initial colonization. Moreover, the motility of bacteria has been shown to mediate dense colonization and severe pathological outcomes in patients [88,89]. Therefore, identifying compounds that target the adhesins or motility of pathogens without inhibiting or killing the growth of bacteria (thus limiting selective pressure to promote resistance development) could be an alternative strategy for the treatment of bacterial infection [90,91].

The surface protein sortase A (SrtA) has been shown to involve in the bacterial biofilm formation and adhesion of several pathogens. However, SrtA is not required for Gram-positive bacterial growth or viability [92,93,94]. SrtA is conveniently located on the cell membrane making it easily accessible to its inhibitors making SrtA an appealing target for anti-virulence drug development. Moreover, mutation of SrtA in *S. aureus* and other Gram-positive bacteria reduces virulence compared with the wild-type strains [95]. Quercitrin (QEN), a natural bioflavonoid isolated from *Sabina pingii var. wilsonii*, is able to remarkably inhibit the enzymatic activity of purified SrtA (Figure 1E) (Table 1) [39]. Moreover, wild-type *S. aureus* Newman treated with QEN showed a significant reduce for the attachment of bacteria to fibronectin-coated or fibrinogen-coated surfaces [39]. QEN also displays the ability to inhibit the biofilm formation of *Streptococcus pneumoniae* by affecting sialic production [40]. However, the in vivo anti-virulence activity of QEN is still unknown.

*Campylobacter jejuni* is one of the leading causes of food poisoning worldwide [96]. Motility offered by flagella of *C. jejuni* has been demonstrated positively involved in the bacterial initial colonization [97,98,99]. *C. jejuni* flagella filament is composed of two flagellin proteins, FlaA and FlaB. The flagellins are synthesized and post-translationally modified by O-linked glycosylation with a nine carbon pseudaminic acid sugar derivative (Pse) that resembles sialic acid at ten and seven amino acids, respectively [100]. Six enzymes in order, PseB, PseC, PseH, PseG, PseI, and PseF are involved in the synthesis of Pse in *C. jejuni* [100]. Motility has been shown to play a critical role in *C. jejuni* pathogenicity, therefore, the Pse synthesis pathway enzymes are considered promising targets for the development of alternative therapeutics. Menard et al. reported screening small-molecule inhibitors for Pse biosynthetic enzymes by using the combination of high-throughput screening and in silico screening [91]. The results found three inhibitors, CD24868, CD26839, and CD36508, effectively inhibit *C. jejuni* flagellin production in a dose-dependent manner (Figure 1E) [91]. 

To deprive pathogens of the virulence factors such as adhesins and motility, which are the cause of the colonization and invasion of host tissues, could modulate the bacterial pathogenesis and in this way the virulence-attenuated bacteria may be defeated by the host immune system. Therefore, the efficiency of flagella- or adhesion-specific monoclonal antibodies to block the bacteria infection and induce immune response to kill pathogens in vivo is worth investigating. 

## 8. Anti-Toxins and Secretion System

Secreted toxins play a major role in the pathogenesis of many bacterial pathogens [101]. Therefore, several toxins have been targeted with the aim of blocking fall into 2 categories: chemical inhibitors and anti-toxin antibodies. Anthrax disease is caused by anthrax toxin secreted by the spore-forming *Bacillus anthracis*. Anthrax toxin is composed of three subunits, including protective antigen (PA); lethal factor (LF), a protease; and edema factor (EF), an adenylate cyclase, which assemble together in binary combinations to form lethal toxin and edema toxin [102,103]. PA binds to cellular receptors, TEM8 or CMG2, and then translocases LF and EF into targeted cells [102,103]. These toxins alter cell signaling pathways in the cytosol, interfere with innate immune responses in early infection stage, and then to induce vascular collapse at late stage [102,103].

The most clinically advanced of antitoxin antibodies is raxibacumab approved by the US Food and Drug Administration in 2012. Raxibacumab is a fully humanized immunoglobulin G1 (IgG1) monoclonal antibody that prevents anthrax toxin binding to its host cell receptor (Figure 1F) [104,105,106]. It is illegal and unethical to perform prophylactic and therapeutic efficacy studies in humans to inform decisions regarding the optimal timing of raxibacumab administration for clinical treatment of anthrax, therefore, raxibacumab is approved on the basis of efficacy in animals [107,108]. Raxibacumab is approved for the treatment of adult and pediatric patients with inhalational anthrax in combination with appropriate antibiotics and for prophylaxis of inhalational anthrax when alternative therapies are not available or not appropriate [107,108]. 

Botulism occurs in infants and adults and is caused by botulinum neurotoxin (BoNT), the most deadly toxin known, secreted by *Clostridium botulinum* [109]. BoNTs are classified into seven serotypes distinguishable with animal antisera and designated from A to G [110]. To identify antibodies which were able to efficiently neutralize BoNT subtype E and F, yeast-displayed single chain Fv (scFv) antibody libraries were constructed from the variable region genes of the heavy (V_H_) and light chains (Vk) of human volunteers immunized with pentavalent BoNT neurotoxoid and tested in a mouse neutralization assay [111,112]. However, the antitoxins used to treat BoNT neutralize circulating toxins but cannot bind or neutralize BoNT that has entered the neuron. Therefore, modifying the structure of *Botulinum* antitoxins to increase their permeability to neurons is worth investigating in the future.

## 9. Engineered Bacteriophage

The potential of employing lytic bacteriophage to act as antimicrobial agents against MDR bacteria has been studied extensively [113,114,115,116,117]. Phage therapy, as it is often referred to, has been used successfully for over 100 years to treat children suffering from severe dysentery [118]. Even after the discovery of antimicrobial agents, phage therapy continues to be used in Eastern Europe and Russia [119]. In order to identified lytic bacteriophages for the treatment of patients infected with XDR *A. baumannii* strains, Leshkasheli et al. isolated two phage, vB_AbaM_3054 and vB_AbaM_3090, through classical amplification from samples of wastewater [118]. Mice were intraperitoneally (i.p.) infected with the XDR *A. baumannii* to cause bacteraemia, then treated with vbB_AbaM_3054 and vB_AbaM_3090 i.p. alone or in combination 2 h after bacterial challenge to evaluate the efficiency of phage-based treatments (6 mice/group). All untreated mice died at day 1, in contrast, phage treatments led to improved survival at day 7 (>80%) [118]. However, the effect was not significantly different between single phage treatments and the combination of both phages. While the development of these phage for clinical applications are ongoing the current results demonstrated that phage-based treatments are high efficacy in mice compared with the untreated controls [118]. 

Dissanayake et al. revealed that a multi-target phage-cocktail significantly reduced the levels of an enterohemorrhagic *E. coli* O157:H7 strain in infected mice and the effectiveness was approximately the same at that observed with ampicillin treatment [120]. Importantly, the phage-cocktail left no detectable impact on the normal gut microbiota composition compared to ampicillin treatment which significantly disrupted the gut microbiota. Moreover, the phage-cocktail treatment had no deleterious impact on the weight of mice; in contrast, remarkable weight loss was observed in the antibiotic treatment group [120]. The ampicillin-treated group showed the greatest weight loss with a 5.44% reduction for post-infection day 1 and continued to show the greatest weight loss compared to all the other groups for days 2, 3, and 5 [120].

Although many studies have reported on the safety and potential of phage therapy for treating patients having bacterial infections [116,117,119,121,122,123], the clinical use of phage therapy is currently awaiting approval in many countries [124]. There are additional issues that must be addressed before bringing phage therapy into the clinic, these include: rapid pathogen resistance to phage after phage treatment [125,126,127,128]; a limited range of target bacterial species/strains [129]; unknown immunogenicity of phage therapy leading to unexpected outcomes [130]; and the inadvertent spread of antibiotic resistant determinants through phage expediting antibiotic resistance [131]. Since the conception of using phages on foods and patients, a substantial number of research reports have described the use of phages to target a variety of bacterial pathogens in food industry or clinical practice. Though some challenges remain, phage biocontrol and therapy are increasingly recognized as an attractive interest and have great potential in our arsenal of tools for safely and naturally eliminating pathogenic bacteria. Moreover, phages are considered as nature’s antibiotics and may also be beneficial for other uses such as a surgical and hospital disinfectant, but have yet to be fully exploited.

## 10. Modulate the Microbiome

Understanding the importance of the microbiome to human health and disease has grown exponentially in the past decade [132,133,134]. In United States, *Clostridium difficile* is the leading cause of infective diarrhea in health-care centers. The pathogenesis of *C. difficile* is influenced by host immune system, antimicrobial treatment history, bacterial toxins (toxin A and toxin B), and host microbiota and its metabolites [135]. *C. difficile* replicates in the colon after the diversity of the beneficial and protective gut microbiota have been disrupted by antimicrobial treatment leading to an altered microbiota. Upon replication to high numbers, the A/B toxins trigger host cellular responses to cause diarrhea, inflammation and tissue damage [135].

The most successful use of the microbiome manipulation as a therapy is the use of fecal microbiota transplant (FMT) treatment to fight recurrent *C. difficile* infections (CDI) [136]. Recently, human stool used for FMT has been classified as an approved biological agent for treatment of CDI by the FDA. Successful FMT treatment rates are very high, and the introduced microbiota appears to be stable in the host for several months [137,138,139]. Moreover, Ianiro et al. have recently compared the efficacy of different FMT protocols, and the results showed that efficacy rates of FMT achieved by all types of protocols were 93% overall (analyzing 15 studies, 1150 subjects). Multiple FMT infusion protocols showed increased efficacy rates in more severe cases compared to single infusion protocols (93% vs. 76%) [140]. However, many questions remain regarding how to optimize FMT treatment. For example, how does the donor’s microbiota influence the health of the recipient after recovery from CDI? Should we monitor the long-term clinical efficacy of the recipient? What is the best formula for the therapeutic microbiota? should there be personalized FTM? How can we maintain a healthy microbiota in the host to avoid the recurrent infection? Can microbiome therapy be used for the treatment of patients infected with MDR-bacteria other than CDI?

Beside to FMT, probiotics are used to alter the microbiome and thus prevent or eradicate infection. Several studies indicate that *Lactobacillus* species and *Saccharomyces boulardii* can efficiently reduce the risk of CDI and antibiotic-associated diarrhea [141]. Another example of biological agent treatment is using antagonistic microorganisms to compete for sites with periodontal disease-causing bacteria in the oral cavity [142]. The results showed that the adhesion of *Porphyromonas gingivalis* ATCC 33277 was inhibited by its antagonistic strains of *Actinomyces naeslundii*, *Haemophilus parainfluenzae*, *Streptococcus mitis*, and *Streptococcus sanguinis* (at least 1.6 cells per adhering antagonist) [142]. 

The human body contains a complex population of bacteria with huge diversity of different species. Bacteria and the chemicals they produce affect the body homeostasis and these effects can have both positive and negative impacts on human health [143,144,145]. The use of healthy human donor flora implanted into the recipient patients appears to be the most complete probiotic treatment available today. It acts as a “broad-spectrum antibiotic” capable of eradicating pathogens and their spores by competition for replicative niches, thus re-balance the homeostasis of the body and microbiota. 

## 11. Conclusions

Antibiotic resistance is dramatically increased worldwide in the past decades and thus these superbugs bring us to the end of current “antibiotic era”. Importantly, patients infected with drug-resistant bacteria are usually at high risk of worse clinical outcomes. Continual epidemiologic surveillance and monitoring of antibiotic prescription and their consumption could delay the spread of antibiotic-resistant microorganisms. In addition, the other potential ways to reduce the rate of emerging resistance is to use combinations of antibiotics or development of alternative therapies. Therefore, concentrating on the discovery of novel compounds that target bacterial systems, such as signal transduction, regulators, virulence, or permeabilization the bacterial membrane, is urgent (Figure 1). Most alternative therapies cannot directly kill the bacteria by disrupting the process of bacterial pathogenesis, however, virulence-attenuated bacteria may efficaciously be defeated by the host immune system or antibiotics. In conclusion, microbiome manipulation and combination therapies are current promising alternative therapies for bacterial infection. However, the discovery of novel classes of antibiotics or alternative therapies requires urgent attention as we march towards a post antibiotic era.

## Figures and Tables

**Figure 1 ijms-21-01061-f001:**
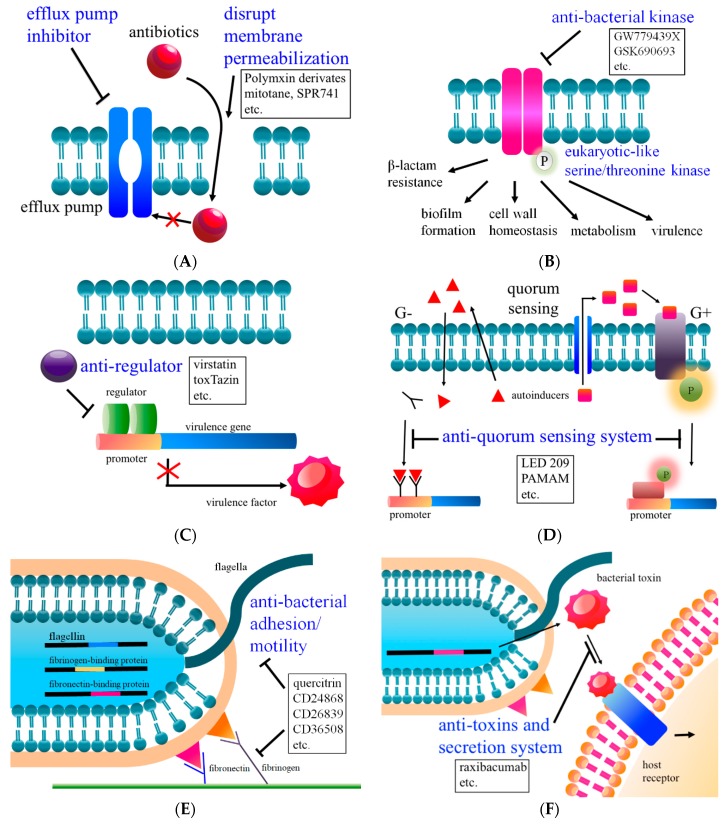
Illustration of alternative antibacterial therapies: (**A**) increase membrane permeability, (**B**) inhibit kinase activity and intrinsic resistance, (**C**) anti-regulators, (**D**) anti-quorum sensing system, (**E**) reduce adhesion and motility, and (**F**) anti-toxins and secretion system.

**Table 1 ijms-21-01061-t001:** Molecular characteristics, action, and function of compounds against bacterial infections.

Compound	Structure and Molecular Weight	Action	Function	Reference
Mitotane	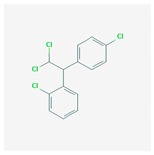 320 g/mol	Unknown	Combine with polymyxin B to target carbapenem-resistant *P. aeruginosa*, *A. baumannii*, and *K. pneumoniae*	[24]
SPR741	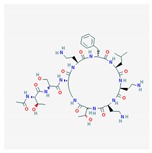 992.1 g/mol	Disrupts the bacterial outer membrane, permeabilizing it to antibiotics	Combine with antibiotics to target antibiotic-resistant *E. coli*, *K. pneumoniae*, and *A. baumannii*	[25,26]
Phenylalanyl arginyl β-naphthylamide (PAβN)	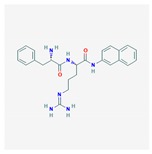 446.5 g/mol	Inhibits Gram-negative efflux pumps	Combine with antibiotics to target fluoroquinolones- resistant *P. aeruginosa*	[31,32]
Curcumin	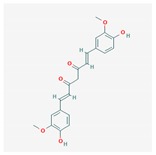 368.4 g/mol	Efflux pump inhibitor	Combine with antibiotics target multidrug- resistant *P. aeruginosa*	[33]
Quinazoline compounds-GI261520A	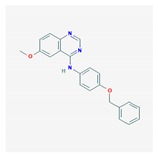 357.4 g/mol	Inhibits kinase activity of PhoP/PhoQ two-component system	Anti-virulence effect by blocking *S.* Typhimurium intramacrophage replication	[34]
GW779439X	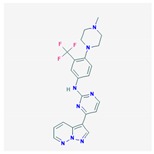 454.5 g/mol	Inhibits *S. aureus* PASTA kinase Stk1 activity	Resensitizes methicillin-resistant *S. aureus* to various β-lactams	[35]
Virstatin	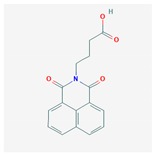 283.28 g/mol	Inhibits dimerization of regulator ToxT of *V. cholerae*	Reduces the nization of *V. cholera* in a murine model of infection	[36,37]
LED209	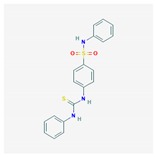 383.5 g/mol	Anti-*E. coli* quorum sensing system	Abolishes EHEC A/E lesion formation and decreases expression of shiga toxin	[38]
Quercitrin	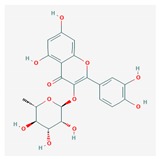 448.4 g/mol	Inhibits the enzymatic activity of surface protein sortase A (SrtA)	Reduces attachment of *S. aureus* to fibronectin-coated or fibrinogen-coated surface; inhibits biofilm formation of *S. pneumoniae*	[39,40]

The structure of compounds were adopted from PubChem database (https://pubchem.ncbi.nlm.nih.gov/).

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
