# Peer review of "Defeating Antibiotic-Resistant Bacteria: Exploring Alternative Therapies for a Post-Antibiotic Era"

_ijms, 2020, doi:10.3390/ijms21031061_

Round 1
Reviewer 1 Report
In recent years, there are urgent needs towards development of novel class of antibiotics and novel approaches for tackling antimicrobial resistance (AMR). This review introduces some alternative targets and therapies, and sounds adequate for publication of this journal. However, I feel that this paper needs to be improved in order to attract more attention from readers of this journal.
1) There are some typos and grammatical errors in this text. The author should check this manuscript carefully.
2) The author should cite the report as follows: O’Neill J, Tackling Drug-Resistant Infections Globally: Final Report and Recommendations. London, UK: Review on Antimicrobial Resistance; 2016; 1-84.
3) Regarding compounds introduced in this review, the chemical structure should be shown as Figures in the revised text. Also, necessary information on compounds should be summarized as Tables in the revised text.
4) Which targets and therapies are particularly promising? The authors should add any sentences about this point in the “Conclusions”.
5) As the authors state the possibility in the latter section, the utilization of human microbiome may be one attractive option in future therapy for infectious diseases. Based on the concept of this paper described in the “Introduction”, it will be more favorable if the authors can introduce some examples related to AMR.
Author Response
1) There are some typos and grammatical errors in this text. The author should check this manuscript carefully.
Response: We ask a native English speaker to revise the manuscript.
2) The author should cite the report as follows: O’Neill J, Tackling Drug-Resistant Infections Globally: Final Report and Recommendations. London, UK: Review on Antimicrobial Resistance; 2016; 1-84.
Response: Add as suggestion (ref 22).
3) Regarding compounds introduced in this review, the chemical structure should be shown as Figures in the revised text. Also, necessary information on compounds should be summarized as Tables in the revised text.
Response: We add the Table 1 to summarize the characteristics of compounds.
4) Which targets and therapies are particularly promising? The authors should add any sentences about this point in the “Conclusions”.
Response: We add the description in the conclusion.
5) As the authors state the possibility in the latter section, the utilization of human microbiome may be one attractive option in future therapy for infectious diseases. Based on the concept of this paper described in the “Introduction”, it will be more favorable if the authors can introduce some examples related to AMR.
Response: We add the description in the introduction and related references (ref 5, 15 and 16).

Reviewer 2 Report
The authors of the work focused on alternative treatment options for bacterial infections. We are aware that resistance to known antibiotics is a significant problem. Unfortunately, we cannot expect new antibiotics that could be used to treat resistant bacterial infections. The authors presented the possibility of controlling infections by using preparations/targets/therapies that do not have antibiotic properties, but e.g. weaken virulence, reduce adhesion, neutralize toxins and so on. It is very interesting and worthwhile. In my opinion, the limitation of this work is the lack of information about which of the described ways of controlling infections is possible to use in the future. The reader can expect an answer if systematic observations have been made in infected patients. Whether advanced in vitro tests or clinical trials were performed. Summarizing each alternative treatment, the authors write: "it is worth investigating", "is still unclear/unknown", "remains to be investigated". This work should have a larger practical aspect. Information about the real chance of using some of the described options in the treatment of infections in patients in the near future would be a value.
Author Response
We thank for the reviewer's comments and revise the manuscript as suggestion.
